# Application of the Regression-Augmented Regionalization Approach for BTOP Model in Ungauged Basins

**Ying Zhu** [1], **Lingxue Liu** [2], **Fangling Qin** [1] , **Li Zhou** [1,3,*] , **Xing Zhang** [4], **Ting Chen** [5], **Xiaodong Li** [1] **and Tianqi Ao** [1,2,*]

1   State Key Laboratory of Hydraulics and Mountain River Engineering, College of Water Resource & Hydropower, Sichuan University, No. 24 South Section 1, Yihuan Road, Chengdu 610065, China; zy197615_abc@163.com (Y.Z.); qinfangling123@163.com (F.Q.); lxdscu@163.com (X.L.)
2   Institute for Disaster Management and Reconstruction, Sichuan University-Hong Kong Polytechnic University, Chengdu 610065, China; liulingxue3@163.com
3   Nanjing Hydraulic Research Institute, Nanjing 210029, China
4   Sichuan Water Resources and Hydroelectric Investigation & Design Institute, SWHI, Chengdu 610072, China; zhangxingscu@163.com
5   Sichuan Meteorological Service Center, Chengdu 610072, China; cting43@foxmail.com
*   Correspondence: zhouli.scu@gmail.com (L.Z.); aotianqi@scu.edu.cn (T.A.)

**Abstract:** Ten years after the Predictions in Ungauged Basins (PUB) initiative was put forward, known as the post-PUB era (2013 onwards), reducing uncertainty in hydrological prediction in ungauged basins still receives considerable attention. This integration or optimization of the traditional regionalization approaches is an effective way to improve the river discharge simulation in the ungauged basins. In the Jialing River, southwest of China, the regression equations of hydrological model parameters and watershed characteristic factors were firstly established, based on the block-wise use of TOPMODEL (BTOP). This paper explored the application of twelve regionalization approaches that were combined with the spatial proximity, physical similarity, integration similarity, and regression-augmented approach in five ungauged target basins. The results showed that the spatial proximity approach performs best in the river discharge simulation of the studied basins, while the regression-augmented regionalization approach is satisfactory as well, indicating a good potential for the application in ungauged basins. However, for the regression-augmented approach, the number of watershed characteristic factors considered in the regression equation impacts the simulated effect, implying that the determination of optimal watershed characteristic factors set by the model parameter regression equation is a crux for the regression-augmented approach, and the regression strength may also be an influencing factor. These findings provide meaningful information to establish a parametric transfer equation, as well as references for the application in data-sparse regions for the BTOP model. Future research should address the classification of the donor basins under the spatial distance between the reference basin and the target basin, and build regression equations of model parameters adopted to regression-augmented regionalization in each classification group, to further explore this approach's potential.

**Keywords:** BTOP model; ungauged basins; river discharge simulation; regionalization approaches; regression-augmented

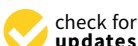



## 1. Introduction

An ungauged basin is a relative concept, which mainly refers to a basin lacking high temporal and spatial resolution (higher observing site density), long historical records, and continuous ground precipitation and river flow data [1,2]. From a global perspective, due to the limited number of hydrological and meteorological observations, plenty of ungauged basins are widely distributed worldwide, especially in developing countries. Small and medium-sized basins are usually defined as basins with a catchment area of

50–3000 km$^2$ [3]. Compared with large basins, these small and medium-sized basins are more common and typical ungauged basins, due to natural conditions or social and economic constraints [1,4,5]. Moreover, discharge simulation is an essential part of water resources management [6], water environment protection [7] and risk management [8]. Therefore, it is a significant challenge for hydrologists to simulate discharge in small and medium watersheds with insufficient data.

In 2003, the International Association for Hydrological Sciences (IAHS) issued a hydrological ten-year plan called the Prediction in Ungauged Basins (PUB) for the next ten years (2003–2012) at the 23rd International Union of Geophysics and Geodesy. It aims to vigorously promote the development of hydrological forecasting in ungauged basins [1,4,9]. The implementation of this project has promoted the development of parametric regionalization approaches for ungauged basins [5,10,11]. Among them, there are three regionalization approaches widely used in river discharge simulation with relatively stable simulation results: regression approach [11–13], spatial proximity approach (SP) [11,14,15], and physical similarity approach (PS) [1,16,17]. The regression approach is the most popular approach in regionalization, and usually utilizes hydrological reference information of the basin to establish the regression equation between model parameters and watershed characteristic factors, such as basin area, the annual average rainfall, and so on [5,11]. SP is based on the climate, and underlying surface conditions change uniformly in the space between the donor basins and the target one [18]. At present, there have been many improvements to the SP, such as the use of inverse distance weighting (IDW) and Kriging for parameter interpolation in the participating basins [19,20]. PS was founded on the existence of similar watershed attributes among basins. For example, Oudin et al. (2008) [11] proposed an integration similarity approach (SP − PS), which takes the spatial distance as an attribute, showing certain advantages over the PS in the 913 studied watersheds in France.

Although nearly ten years have passed since the end of PUB in 2012, the research on regionalization approaches for river discharge simulation in the ungauged basin has never stopped. Arsenault and Brissette (2014) [13] introduced the regionalization approach of regression-augmented (RA) for the first time, essentially combining the multiple linear regression approach with the SP and PS approach, and found RA outperformed SP and PS regionalization approaches in 268 Canadian watersheds. Hong et al. (2017) [21] carried out spatial interpolation for a runoff simulation in the ungauged basin, based on the information diffusion model of the genetic algorithm in the Yellow River Basin. The results showed SP combined with a machine learning method has better simulation results than the SP combined with the IDW method or Cokriging method. In Tyrol, Pugliese et al. (2018) [22] simulated rainfall-runoff on a macro-scale. The results showed that the SP combined with the data assimilation method could significantly improve the runoff prediction of the basin with very low station density (one gauge per 2000 km$^2$). Arsenault et al. (2019) [18] used six regionalization approaches (SP, SP-IDW, SP-IDW-RA, PS, PS-IDW, PS-IDW-RA) to predict streamflow, based on three hydrological models in ungauged catchments in Mexico. The results showed that the SP approach performed best, and the regression-augmented spatial similarity approach was also a great choice in certain conditions. Kanishka and Tieldho (2020) [23] pointed out that before utilizing regionalization approaches for river discharge simulation in an ungauged basin, isomap and principal component analysis could be used to classify the reference basins. This kind of procedure can improve the result of river discharge simulation in the target basin.

More studies have shown that optimizing and combining the spatial proximity approach, physical similarity approach, or regression approach is a favorable method to improve the accuracy of simulation results of the regionalization approach [13,18,21–23]. However, it is difficult to determine the best proper regionalization approach, due to various factors, such as the heterogeneity of the basin surface condition and the diversity of hydrological models [1]. As a semi-physically distributed hydrological model, the block-wise use of TOPMODEL (BTOP) has been applied in many watersheds in the world [24–31]. The model has fewer tuning parameters, and each parameter holds the physical interpreta-

tion. Therefore, it has the potential for its parameters to be related to basin features, and can show superiority in the parameter transfer functions in the ungauged basins [32]. However, limited research with inadequate data is related to regionalization approaches based on the BTOP, to the authors' best knowledge. Ao et al. (2006) [32] linked the parameters of the BTOP model with the physical features of the watershed through multiple regression equations, to explore the possibility of establishing an improved physically-based a priori parameter estimation technique. They found that 'improved' a priori parameter estimation techniques have the potential to reduce parameter uncertainty and enable prediction in ungauged basins. Huang et al. (2015) [33] attempted to build a simple linear relationship between BTOP sensitive parameters and possible watershed features. They defined two new characteristics (mean soil particle size and mean root depth) based on the model input data, showing that the newly defined characteristics played an essential role in building the relationship between the stability of parameters and watershed features. However, none of the above research specifically studied the regionalization approaches for the BTOP model in the ungauged basin. Accordingly, this paper is committed to filling the gap.

　　　This study explores whether the RA regionalization approach could improve discharge simulation, compared with the traditional regionalization approach. We selected 11 small and medium basins in the middle and lower reaches of the Jialing River, the largest tributary of the Yangtze River in the southwest of China. Then, a total of 12 regionalization approaches, which combined with the SP, PS, SP − PS, and the regression approach, are applied to the study area to facilitate the BTOP model in obtaining river discharge simulation results. Finally, an evaluation is conducted to compare the RA regionalization approach with the traditional regionalization approach.

## 2. Materials

### 2.1. Study Area

　　　Jialing River is an important tributary of the Yangtze River, and is an essential source of flood water and sediments for the Three Gorges Reservoir [34–36]. We selected 11 small and medium basins in the middle and lower reaches of the Jialing River basin with similar distance and discharge observation data of sub-basins as the study area (Figure 1). The selected basins are located in mountainous regions, and can be considered a typical small and medium-sized ungauged basin, due to sparse gauges density.

　　　As shown in Figure 1, the studied catchments are located between 30–32° N and 105–109° E, and each catchment area is monitored by a river gauge. The study basins belong to a subtropical monsoon humid climate, with an annual average rainfall of about 1000 mm and an annual average temperature range of 14.7–17.6 °C. Except for MB(ID6), which is located at the edge of the Sichuan Basin and has a high altitude and low temperature, there is little difference in other basins. Most of the basins in the study area have a relatively small mean slope, with a range of 3.93~23.60°. Study area has similar spatial distance and hydrological characteristics. The 11 basins selected in this study can be considered as a typical ungauged mountainous catchments, which have a great reference significance for the discharge simulation of ungauged basins worldwide.

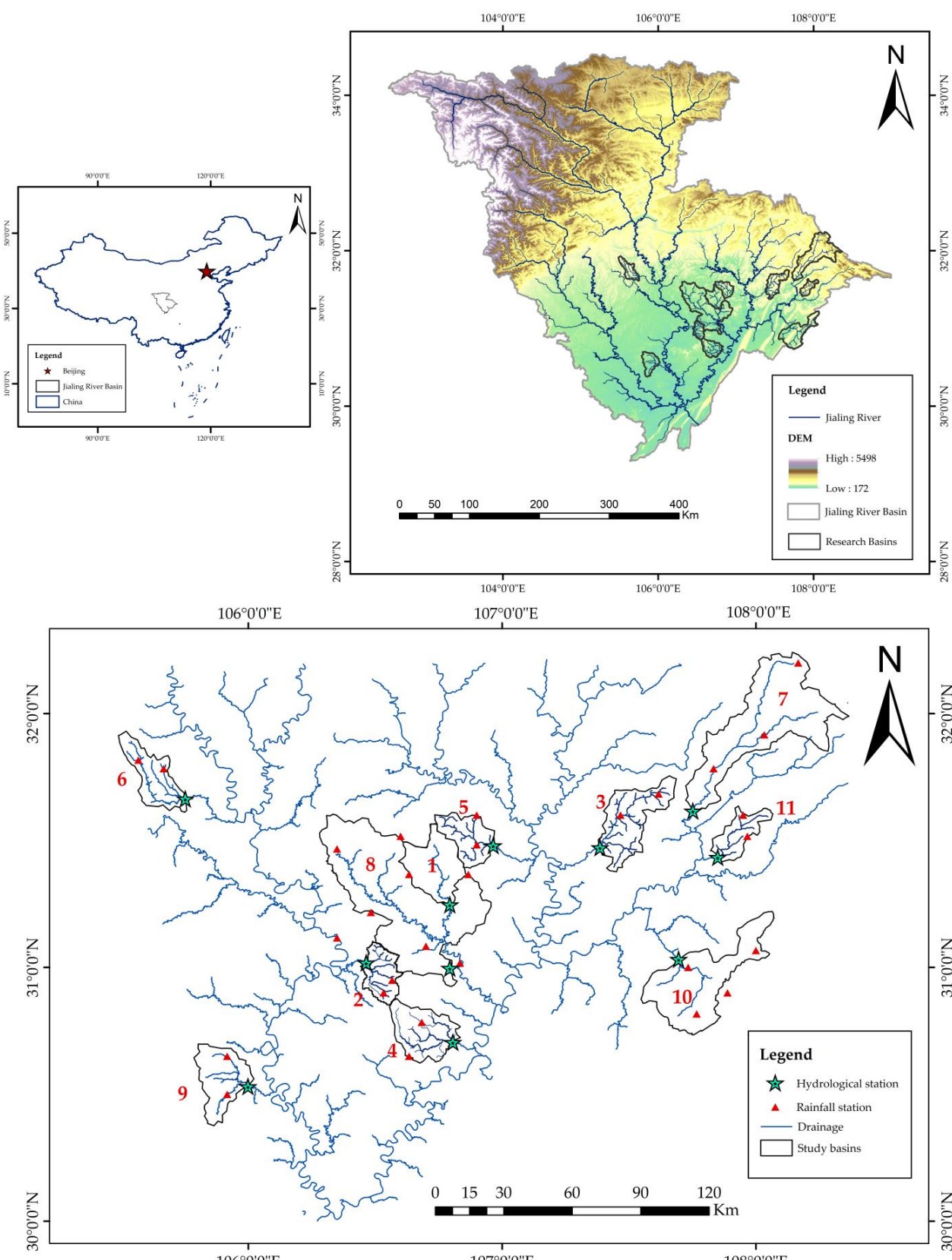

**Figure 1.** Location, ground observation, and DEM of the 11 catchments in the study area. Abbreviations and the identification number for each study basin: Shuixiazi is SXZ (ID1); Boyang is BY (ID2); Yonghong is YH (ID3); Changtanqiao is CTQ (ID4); Yuantuo is YT(ID5); Hongyan is HY(ID6); Maoba is MB (ID7); Jingbian is JB (ID8); Zhaojiaci is ZJC (ID9); Mingyuetan is MYT (ID10); Qingxi is QX (ID11).

*2.2. BTOP Model*

The BTOP model, a semi-distributed hydrological model, is designed to perform simulations of rainfall-runoff, soil moisture, unsaturated and saturated groundwater, and river discharge processes at each BTOP grid in watersheds [37,38]. Moreover, BTOP is referred to as BTOPMC when the Muskingum-Cunge (MC) method is applied for the flow routing phase [32,39]. As a significant part of the Yamanashi Hydrological Model (YHyM) system for river basin simulation, BTOP has been applied to many river basins worldwide, especially in warm, humid catchments [25,33,40,41]. The current version of the BTOP model has five calibrated parameters of $n_{0c}$, $m$, $\alpha$, $SD_{ba}$r, and $D_o$ (which is calculated by the fraction and dischargebility of sand, silt, and clay). Table 1 summarizes a range of parameter values with the default BTOP model parameters obtained for the Mekong River basin [24,38].

**Table 1.** Model parameters of BTOP.

| Zone | | Model Parameters | Value of Parameters | | | Unit |
|---|---|---|---|---|---|---|
| | | | Default | Lowest | Highest | |
| Flow generation | Root | Block-wise drying parameter ($\alpha$) | −6 | −10 | 10 | \ |
| | Unsaturated/Saturated | Decay factor ($m$) | 0.057 | 0.01 | 0.1 | m |
| | Unsaturated | Block-average saturation deficit ($SD_{bar}$) | 0.048374 | 0.001 | 0.9 | m |
| | Saturated | Groundwater dischargeability of the sand ($D_{0sand}$) | 0.09 | 0.01 | 2.0 | m/h |
| | | Groundwater dischargeability of the silt ($D_{0silt}$) | 0.04 | 0.01 | 2.0 | |
| | | Groundwater dischargeability of the clay ($D_{0clay}$) | 0.01 | 0.01 | 2.0 | |
| Flow routing | Channel | Block average Manning's coefficient ($n_{0c}$) | 0.0006 | 0.00001 | 0.8 | s/m$^{1/3}$ |

Compared with other conceptual hydrological models, the BTOP model parameters have specific physical interpretations. Moreover, the number of tuning parameters is fewer than other physically-based distributed hydrological models, such as SWAT, which means that the parameter interaction and uncertainties are less [32]. Therefore, a robust relationship between the model parameters and the physical characteristics of the river basin is more likely to be established for the BTOP model. In summary, applying the BTOP model to the study of the regionalization approach of the data-sparse watershed has apparent advantages.

2.2.1. Core of BTOPMC

The structure of BTOP includes core components and optional modules. For the runoff simulation section, the BTOP core modules are briefly introduced as follows [33].

(a)  Topographic sub-module

The topographic module of BTOP is based on DEM and soil type data. Topographic processing includes digital river network generation, sub-basin division using river station locations for delineating drainage area boundaries, effective contributing area calculation, and topographic index calculation [42].

(b)  Runoff generation sub-module

The runoff generation in BTOP is calculated in each grid cell. The processing divides into four zones vertically: vegetation, root, unsaturated, and subsurface [38]. There are four model parameters involving calibration and optimization for each BTOP block, which can be less or equal to the number of study basins (see Table 1). Furthermore, the Hortonian overland flow option was selected in our study. The detailed rainfall-runoff calculation of each zone can be referred to in the user manual of BTOP [38].

(c)　　Flow routing sub-module

The flow routing sub-module of BTOP was constructed by the Muskingum-Cunge (MC) method [43,44], but the original MC method could not explain the backwater phenomena. Thus, the modified MC method was developed to overcome this problem [45,46], by solving the inherent negative outflow problem and ensuring the flow routing accuracy of BTOP. The modified MC method also indicated a better performance than the original MC method implemented in earlier applications of BTOP [32,37]. Moreover, at this stage, there is just one model parameter, the block-average Manning's coefficient $n_{0c}$, that needs to be optimized and calibrated, and $n_{0c}$ is used to calculate the Manning's coefficient at each grid using grid slope.

2.2.2. Evaluation Criteria

An appropriate objective function should be used to evaluate the proximity between simulated and measured river discharge. In this study, the coefficient of determination ($R^2$) and Nash–Sutcliffe Efficiency ($NSE$) [47] were selected as the objective function of the model simulation evaluation. The formula of evaluation criteria are as follows:

$$R^2 = [\sum_{i=1}^{n} (Q_i^s - \overline{Q^s})(Q_i^o - \overline{Q^o})]^2 / \sum_{i=1}^{n} (Q_i^s - \overline{Q^s})^2 \cdot \sum_{i=1}^{n} (Q_i^o - \overline{Q^o})^2 \qquad (1)$$

$$NSE = 1 - \sum_{i=1}^{n} (Q_i^s - Q_i^o)^2 / \sum_{i=1}^{n} (Q_i^o - \overline{Q^o})^2 \qquad (2)$$

where, $Q^s$ and $Q^o$ represent the simulated flow rate and the measured flow rate, respectively. The $i$ is the number of time series of discharge, $n$ is time steps in the evaluation period. The value range of $R^2$ is 0~1. The closer $R^2$ is to 1, the more consistent the change process lines between the simulated value and the measured value. The value range of $NSE$ is $-\infty$–1. The closer $NSE$ is to 1, the better is the simulation efficiency [47].

*2.3. Data*

We collected 37 rainfall stations and 11 hydrological stations in the study basins. Each basin includes at least three rainfall stations and a hydrological station at the outlet. Daily streamflow records and daily rainfall records from 1981–1987 were derived from hydrologic year books published by the Hydrologic Bureau of the Ministry of Water Resources of China. Other input data of the BTOP model are globally available data: the topography is represented by the DEM with an original resolution of 30 m, obtained from the geospatial data cloud [48]. The land cover dataset on a grid-scale of 500 m was accessed from the USGS Land Cover Institute [49]. Food and Agriculture Organization (FAO) provides the soil map at the scale of 1:5 million [50]. The National Centers for Environmental Information (NCEI) provides the leaf area index (LAI) with an original resolution of 0.05 degree [51]. We employed the potential evapotranspiration from the Climatic Research Unit (CRU), with an original resolution of 0.5 degree [52]; the potential intercept evaporation with an original resolution of 0.25 degree was obtained from the Global Land Evaporation Amsterdam Model (GLEAM) [53]. All the above datasets were resampled to 500 m for BTOP model simulation. The BTOP models were set up for 11 study catchments of Jialing River, and each catchment boundary was controlled by the river hydrological station.

**3. Methodology**

Figure 2 shows the framework of this research. First, the model calibration was carried out for 11 study basins to obtain the parameter calibration values of the hydrological model. Then, each type of model calibrated parameter and each type of watershed attribute characteristic factor were calculated as correlation coefficients to select the influential watershed characteristic factors. After that, the *SPSS* software [54,55] was used to establish the linear regression equation between 5 parameters of the BTOP model and the watershed attribute characteristic factors with a higher correlation coefficient ($R > 0.6$). In the establishment of

the regression equation for model parameters, to explore whether the more consideration of the basin characteristic factors leads to a better model fitting effect, we established a one-variable linear regression equation (regression I), two-variable linear regression equation (regression II), and a multivariate linear regression equation (regression III) for the corresponding model parameters. In regression III, the basin characteristic factors that had an R greater than 0.6 with parameters were considered. The fitting effect of the regression equation was evaluated by $R^2$.

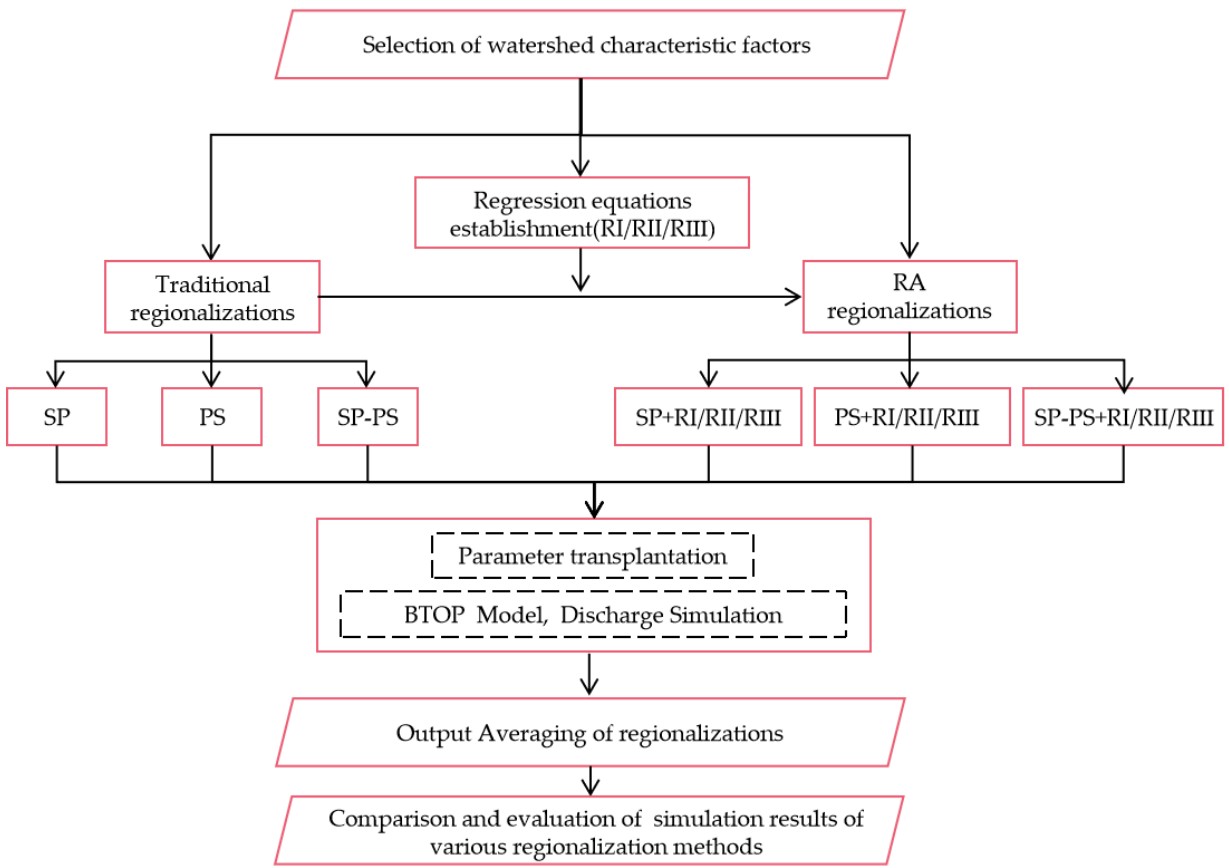

**Figure 2.** The framework of the study. SP, PS, and SP − PS are abbreviations for spatial proximity approach, physical similarity approach, and an integration similarity approach (SP − PS); the R I, RII, and RIII represent regression equations with different numbers of independent variables.

According to the hydrologic year books, in the 1986 daily river discharge data, the YH and MB basins were missing for several days, and there were a few discharge data estimated by hydrological methods in the basins of SXZ, YT, and ZJC. Therefore, those five basins were selected as basins with a lack of river discharge data, and SP, PS, and SP − PS approaches were analyzed in turn. According to the target data-deficient basin, the similarity ranking of gauged basins was carried out, and the appropriate donor basins were selected.

The next step was based on the BTOP model, using traditional regionalization approaches and RA regionalization approaches to simulate discharge in the ungauged basin. The output averaging approach was adopted to deal with the discharge simulation results.

Finally, we compared the evaluation criteria of simulation results of traditional and RA regionalization approaches. The details of the methodology are described in Sections 3.1–3.4.

*3.1. Selection of Watershed Characteristic Factors*

When using the PS and SP − PS to carry out parameter transplantation in the ungauged basins, it involved the selection of basin descriptors. Basin characteristic descriptors selection in the regionalization study of runoff could be classified into two categories: physical descriptors and hydroclimatic descriptors. Basin area ($Ac$, km$^2$), average elevation ($E$, m), average slope ($S$, °), forest cover fraction ($W$, %), and average topographic index ($\lambda$) are usually selected as physical descriptors. The annual average rainfall ($P$, mm) and annual runoff coefficient ($RC$) are generally chosen as the hydroclimatic descriptors [11]. In addition, soil type and land cover type are two essential characteristics of the basin. They are also important influencing factors in the process of rainfall and runoff simulation of hydrological models. According to the simple linear modeling method explored by Huang et al. (2015) [33], two new watershed characteristic factors, mean soil particle size ($\Phi$) and mean root depth ($r$), are employed as factors in this study as well. The $\Phi$ not only introduces the proportion information of each component of soil type but also introduces the ordering information of soil particle size. Its formula is as follows:

$$\phi = \sum_{i=1}^{n} \phi_i \cdot r_i \tag{3}$$

where, $\Phi_i$ is the proportion of $i$ kinds of soil components, $r_i$ refers to the sorting number of the size of the $i$ kinds of soil components (each soil type of BTOP is composed of clay, soil, and sand), $n$ refers to total types of soil components.

Similarly, land-cover types in the BTOP model were classified into 17 categories by IGBP of USGS Land Cover Research Institute. The $r$ introduces information on the proportion of each type of land use type in the watershed and its root depth information:

$$r = \sum_{i=1}^{m} r_i \cdot t_i \tag{4}$$

where, $r_i$ is the root depth corresponding to each land-use type, $t_i$ is the proportion of the area occupied by each land-cover type in the watershed, and $m$ is 17 categories of land-cover in the research watershed.

In summary, nine watershed characteristic factors selected in this study are shown in Table 2.

**Table 2.** Attribute characteristics of research basins.

| Basin Number | River Basin | Calibration Period | Basin Area ($Ac$, km$^2$) | Mean Elevation ($E$, m) | Mean Slope ($S$, °) | Mean Topographic Index ($\lambda$) | Fraction of Forest Cover ($W$, %) | Mean Annual Rainfall ($p$, mm) | Annual Runoff Coefficient ($RC$) | Mean Root Depth ($r$, m) | Mean Soil Particle Size ($\varphi$) |
|---|---|---|---|---|---|---|---|---|---|---|---|
| ID1 | SXZ | 1981–1985 | 487.52 | 475.06 | 9.74 | 11.06 | 25.95 | 1310 | 0.42 | 0.77 | 2.08 |
| ID2 | BY | 1981–1985 | 274.89 | 401.43 | 6.97 | 11.12 | 8.15 | 1068 | 0.45 | 0.74 | 2.04 |
| ID3 | YH | 1983–1985 | 553.1 | 657.25 | 13.11 | 9.84 | 46.85 | 1259 | 0.52 | 0.88 | 2.01 |
| ID4 | CTQ | 1981–1985 | 458.74 | 404.34 | 3.93 | 11.57 | 11.05 | 1058 | 0.45 | 0.76 | 2.05 |
| ID5 | YT | 1981–1985 | 322.19 | 520.58 | 11.87 | 10.57 | 22.87 | 1218 | 0.49 | 0.74 | 2.04 |
| ID6 | HY | 1981–1985 | 382.29 | 610.57 | 11.21 | 10.38 | 16.05 | 1053 | 0.3 | 0.73 | 2.04 |
| ID7 | MB | 1981–1985 | 1454.41 | 1036.99 | 23.6 | 9.55 | 67.83 | 1540 | 0.71 | 1.06 | 1.99 |
| ID8 | JB | 1981–1984 | 2539.25 | 440.78 | 8.57 | 11.22 | 13.39 | 1160 | 0.43 | 0.74 | 2.09 |
| ID9 | ZJC | 1981–1985 | 457.39 | 348.79 | 4.13 | 11.61 | 9.49 | 959 | 0.36 | 0.75 | 2.05 |
| ID10 | MYT | 1981–1983 | 845.33 | 532.69 | 8.94 | 11.36 | 42.84 | 1323 | 0.54 | 0.95 | 2.02 |
| ID11 | QX | 1984–1985 | 249.75 | 952.47 | 16.45 | 9.57 | 55.05 | 1604 | 0.54 | 1.09 | 2.00 |

*3.2. Traditional Regionalization*

3.2.1. Spatial Proximity Approach (SP)

The basic principle of the SP approach is that the climate and underlying surface conditions change uniformly in space, and it is believed that similar fields have similar hydrological behaviors [18]. In this study, the centroid coordinates of each watershed was calculated by *ArcGIS* software, and the spatial distance value from each participating basin to the target basin was obtained, respectively.

### 3.2.2. Physical Similarity Approach (PS)

The PS approach refers to the approach of transmitting hydrological information in a data-deficient basin using reference basins that are similar in terms of basin characteristics. It is usually divided into two categories: basin feature ranking and physical weight methods [11,18]. According to the hydrological characteristic values of each basin, the PS is used to calculate the physical similarity values of each reference basin to the ungauged basin. The smaller the attribute similarity value is, the higher the watershed similarity is [18]. The formula of the physical similarity approach is as follows:

$$x = \sum_{i=1}^{k} \frac{\left| x_i^G - x_i^U \right|}{\Delta x_i} \tag{5}$$

where, $x_i^G$ is the $i$-th watershed characteristic factor of the reference basin; $x_i^U$ is the corresponding watershed characteristic factor of the data-sparse basin; $\Delta x_i$ is the difference between the maximum and minimum value of the $i$-th watershed characteristic factor of the donor basins; and $k$ is the number of selected watershed characteristic factors.

### 3.2.3. Integration Similarity Approach (SP − PS)

Oudin et al. (2008) [11] proposed an integration similarity approach (SP − PS) combining the SP approach and the PS approach in the comparative study of 913 watersheds' regionalization approaches in France in 2008. This approach takes the spatial distance as an attribute. Then, the PS approach formula is used to calculate the similarity value between the donor basin and the target basin.

### 3.3. 'Regression-Augmented' (RA) Regionalization

The regression-augmented (RA) regionalization approach combines the regression-based approach with the SP, PS, and SP − PS approach: the regression equation was used to obtain the relevant model parameters, and the remaining parameters of the model were transplanted according to the reference basins selected by the SP, PS, and SP − PS approach. The regression-based approach establishes regression equations for the model parameters that have a high correlation with watershed attributes. It is generally believed that the correlation coefficient between model parameters and watershed characteristic factors is greater than 0.5, indicating that the established regression equation has application significance [11,18]. Usually, to build the regression equation of model parameters and watershed characteristic factors, only the basins with better simulation performance judged by model calibration results are selected in the study. Then, the corresponding model parameters in the data-deficient area can be estimated based on the regression equations. Thus, each model parameter is independently evaluated in the regression-based approach. However, the model parameter set cannot be evaluated and established simultaneously, which leads to the partial cohesion of the hydrological model parameter set being lost, and affects the simulation performance of the model [11,18]. The RA regionalization approach overcomes the shortcoming of the regression-based approach and makes full use of the watershed attribute characteristics of the ungauged basin and the parameter set of the reference catchment in the parameter transfer process, thus retaining the cohesion of the transferred parameter set [13].

### 3.4. Output Averaging of Regionalization

When the regionalization approach of parameter transplantation is applied to the ungauged basin, the number of reference basins will affect river discharge simulation and the prediction of the target basin [1]. When 2–5 reference basins are used for the regionalization approach, the model simulation efficiency is more satisfactory [56]. In this study, three reference basins were selected when the regional approach was used for parameter transplantation of the ungauged basin. When the traditional regionalization approach uses parameter sets of reference basin to simulate the runoff of the target watershed based on

the hydrological model, there are two approaches to deal with the transplanting process: one is the parameter averaging, and the other is the output averaging [11,13]. Parameter averaging is conducted by taking the average value of the model parameters of the selected reference basins, and then transplanting them to the target basin for river discharge simulation. The output averaging indicates that the calibrated parameter sets of each reference basin are transplanted to the target basin respectively, and then the average river discharge simulation results are taken. Previous studies have shown that some reference watershed information may be lost in the process of parameter averaging, and the simulation results are often not as good as the output averaging [11,13,57]. Moreover, in this study, some parameters of the RA approach were obtained by regression equations, which could not be averaged. After comprehensive consideration, the output averaging approach was adopted to obtain the river discharge simulation results of the target basin in this study.

## 4. Results and Discussions

### 4.1. Model Calibration and Validation

Table 3 shows the calibration and validation results of river discharge simulation using the BTOP model of each basin, and Table 4 shows their calibrated parameters. The *NSE* of the research basin during calibration was basically above 0.60 (except for the QX (ID11), the model performance was generally good, and the model performance of small basins, such as JB (ID8), MB (ID7) and MYT (ID10) were excellent. The $R^2$ of BTOP simulation results were all greater than 0.6, and the hydrographs of river discharge simulation results and the measured hydrographs had a good fit degree. In the validation period, the *NSE* values were all above 0.50, and the $R^2$ values were above 0.60, reflecting good performance of the BTOP model simulation. The *NSE* values of the BY (ID2), JB (ID8), MB (ID7), QX (ID11), SXZ (ID1), and YT (ID5) during the validation period were even higher than that during the calibration period. Since the study basin belongs to the small and medium-sized mountain basins, and in the 1980s, the hydrological and meteorological monitoring had certain limitations, and the rainfall data itself had a large error. Therefore, according to the simulation results in Table 3, the BTOP hydrological model presented a good simulation effect in the daily river discharge simulation of 11 study basins, and the regression equation between the model calibration parameters and the characteristic factors of the basin were reliable.

**Table 3.** Model performance during calibration and validation period of research basins.

| Basin Abbreviation | Simulation Period | SXZ (ID1) | BY (ID2) | YH (ID3) | CTQ (ID4) | YT (ID5) | HY (ID6) | MB (ID7) | JB (ID8) | ZJC (ID9) | MYT (ID10) | QX (ID11) |
|---|---|---|---|---|---|---|---|---|---|---|---|---|
| *NSE* | Calibration | 0.69 | 0.63 | 0.70 | 0.66 | 0.64 | 0.64 | 0.71 | 0.71 | 0.67 | 0.73 | 0.60 |
| $R^2$ | (1981–1985) | 0.71 | 0.64 | 0.73 | 0.66 | 0.64 | 0.64 | 0.71 | 0.72 | 0.68 | 0.76 | 0.63 |
| *NSE* | Validation | 0.71 | 0.65 | 0.55 | 0.61 | 0.71 | 0.52 | 0.71 | 0.72 | 0.57 | 0.60 | 0.61 |
| $R^2$ | (1987) | 0.79 | 0.71 | 0.60 | 0.64 | 0.73 | 0.67 | 0.74 | 0.72 | 0.68 | 0.79 | 0.73 |

**Table 4.** Model calibrated parameters of 11 study basins.

| Basin Number | River Basin | $D_{0clay}$ | $D_{0sand}$ | $D_{0silt}$ | $SD_{bar}$ | $m$ | $n_{0c}$ | $\alpha$ |
|---|---|---|---|---|---|---|---|---|
| ID1 | SXZ | 0.902669 | 0.662340 | 1.063725 | 0.429484 | 0.036628 | 0.025459 | −3.985685 |
| ID2 | BY | 0.111670 | 0.807359 | 0.920748 | 0.343624 | 0.032453 | 0.038815 | −5.102666 |
| ID3 | YH | 0.026699 | 0.025412 | 0.024224 | 0.884511 | 0.040897 | 0.484106 | 7.408702 |
| ID4 | CTQ | 1.032118 | 1.278913 | 0.879630 | 0.487423 | 0.028796 | 0.197 | −7.186764 |
| ID5 | YT | 1.088008 | 1.227803 | 1.035062 | 0.418472 | 0.030564 | 0.228849 | −4.919591 |
| ID6 | HY | 0.591840 | 1.255793 | 0.768696 | 0.733941 | 0.031305 | 0.055943 | 0.447747 |
| ID7 | MB | 0.912159 | 1.475803 | 1.247493 | 0.891302 | 0.050496 | 0.009972 | 7.992502 |
| ID8 | JB | 1.269936 | 1.227124 | 0.868861 | 0.418652 | 0.031874 | 0.026879 | −1.158098 |
| ID9 | ZJC | 0.507032 | 0.339045 | 1.768506 | 0.66618 | 0.029071 | 0.495117 | −2.983849 |
| ID10 | MYT | 0.020146 | 0.020113 | 0.020131 | 0.899949 | 0.046421 | 0.495833 | 7.951358 |
| ID11 | QX | 0.802537 | 0.611314 | 0.578350 | 0.736381 | 0.033526 | 0.02746 | 1.397548 |

### 4.2. Regression Equations Establishment

This study individually analyzed the correlation between the BTOP calibrated parameters of the 11 small and medium basins in Table 4, and the watershed characteristic factors in Table 2. We found that only three parameters ($SD_{bar}$, $m$, and $\alpha$) in the BTOP model had correlation coefficient values higher than 0.6 with watershed characteristic factors. As shown in Figure 3a, the correlation coefficient of parameter $SD_{bar}$ with the mean soil particle size $\varphi$, the forest coverage rate $W$, the average root depth $r$, the average watershed elevation $E$ are $-0.773$, 0.739, 0.696, and 0.620, respectively. The parameter $SD_{bar}$ was determined in the unsaturated zone of the model river discharge simulation, which describes the condition of soil water content [38], and the $\varphi$, $W$, $r$, and $E$ had an apparent impact on the soil water content in the runoff phase. Thus, the research results are reasonable and have physical significance. In Figure 3b, the correlation coefficient between the model parameter $m$ and $W$ was 0.813, the annual runoff coefficient $RC$ was 0.781, and the $r$ was 0.690. The $m$ is the decay factor of the lateral transmittance of the groundwater [33]. Therefore, the research results show that the high correlation between this parameter and $W$ (reflecting land-cover), $r$ (reflecting land-cover), and $RC$ (affected by soil type) can be explained. As for parameter $\alpha$ shown in Figure 3c, the correlation coefficient of $\alpha$ and $W$, $r$, $\varphi$, and $E$ were 0.810, 0.711, $-0.690$, 0.645. The parameter $\alpha$ is the drying coefficient of the root zone, which reflects the drying capacity of the soil surface in the root zone. Therefore, soil type and land use are statistically significant influencing factors.

The multiple linear regression analysis module of the *SPSS* software was used to establish the regression equation between model parameters and basin characteristic factors. According to the results of correlation analysis, each of the three model parameters, $SD_{bar}$, $m$, and $\alpha$, had more than three basin characteristic factors with their correlation coefficient $R > 0.6$. The SPSS fitting results of each regression approach are shown in Table 5, and we can see that the deterministic coefficients of regression I, II, and III increased successively. It indicates that the more watershed characteristic factors were considered, the better the fitting effect between model parameters and selected watershed characteristic factors would be. However, when the regression equation was combined with the traditional regionalization approach, the combination effect of regression III may not be as good as regression I and II. This issue is analyzed in the following two sections.

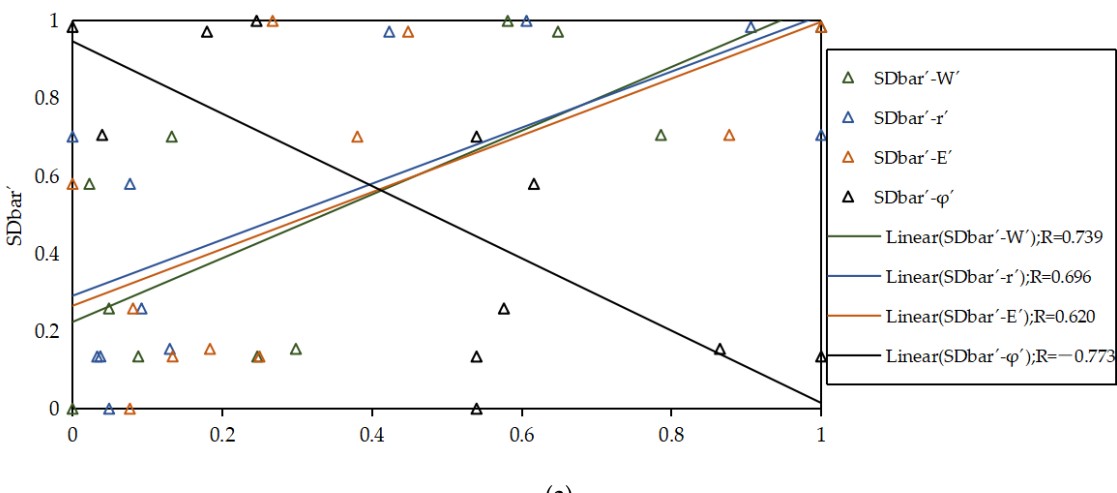

(a)

**Figure 3.** *Cont.*

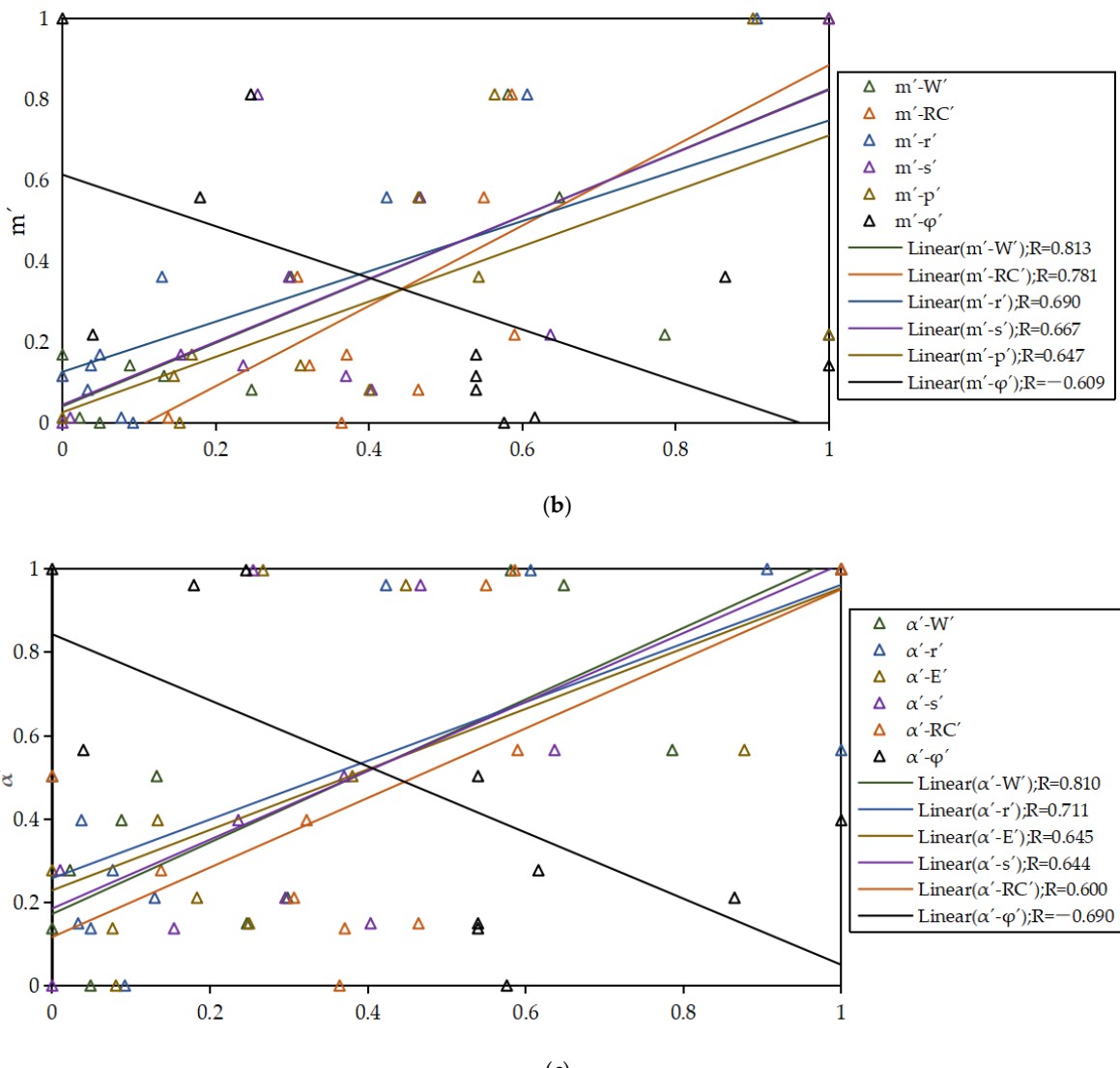

**Figure 3.** Correlation diagram of BTOP model parameters and watershed characteristic factors ($R > 0.6$). All data are normalized by the Min-Max normalization approach. The y-axis represents the normalized value of (**a**) $SD_{bar}'$; (**b**) $m'$; (**c**) $\alpha'$. The x-axis represents the normalized value of watershed characteristic factors.

**Table 5.** Regression equations of model parameters and attribute characteristics of research basins and their fitting evaluation indexes.

| Regression Type | Model Parameters | Analytic Equation | $R^2$ |
|---|---|---|---|
| Regression I (one-variable) | $SD_{bar}$ | $SD_{bar} = 11.379275 - 5.276112 \cdot \varphi$ | 0.598 |
| | $m$ | $m = 0.027348 + 0.000285 \cdot W$ | 0.661 |
| | $\alpha$ | $\alpha = -6.354825 + 0.218348 \cdot W$ | 0.656 |
| Regression II (Two-variable) | $SD_{bar}$ | $SD_{bar} = 7.654843 - 3.495046 \cdot \varphi + 0.003276 \cdot W$ | 0.630 |
| | $m$ | $m = 0.019598 + 0.000188 \cdot W + 0.022371 \cdot RC$ | 0.692 |
| | $\alpha$ | $\alpha = 2.186949 + 0.297397 \cdot W - 12.924569 \cdot r$ | 0.669 |
| Regression III (Multivariate) | $SD_{bar}$ | $SD_{bar} = 9.513535 - 4.228658 \cdot \varphi + 0.007865 \cdot W - 0.358424 \cdot r - 0.000339 \cdot E$ | 0.661 |
| | $m$ | $m = -0.202705 + 0.000641 \cdot W + 0.02875 \cdot RC - 0.000032 \cdot p - 0.00019 \cdot s + 0.000975 \cdot r + 0.121263 \cdot \varphi$ | 0.824 |
| | $\alpha$ | $\alpha = 134.584381 + 0.336402 \cdot W + 14.951869 \cdot r - 63.658863 \cdot \varphi - 0.038991 \cdot E + 0.968231 \cdot s - 31.084738 \cdot RC$ | 0.771 |

### 4.3. Similarity Ranking of Basins

While exploring the effect of each regionalization approach on river discharge simulation of the data-deficient basin, an independent "data-deficient basin" should be selected. According to Figure 1, except for the JB (ID8), which contains an inset sub-basin, SXZ (ID1), all other watersheds were independent catchments. Here, we selected SXZ (ID1), YH (ID3), YT (ID5), MB (ID7), and ZJC (ID9) as the data-deficient basins, which were drained by different tributaries of the Jialing River. The similarity sorting was performed for each data-deficient basin according to the SP, PS, and SP − PS, and are shown in Figure 4. Table 6 displays the three most similar reference basins selected, according to the above three approaches and their similar values to the ungauged basin. The river discharge simulation results of the three reference basins corresponding to the target basin were obtained based on the weight of similarity values in Section 4.4.

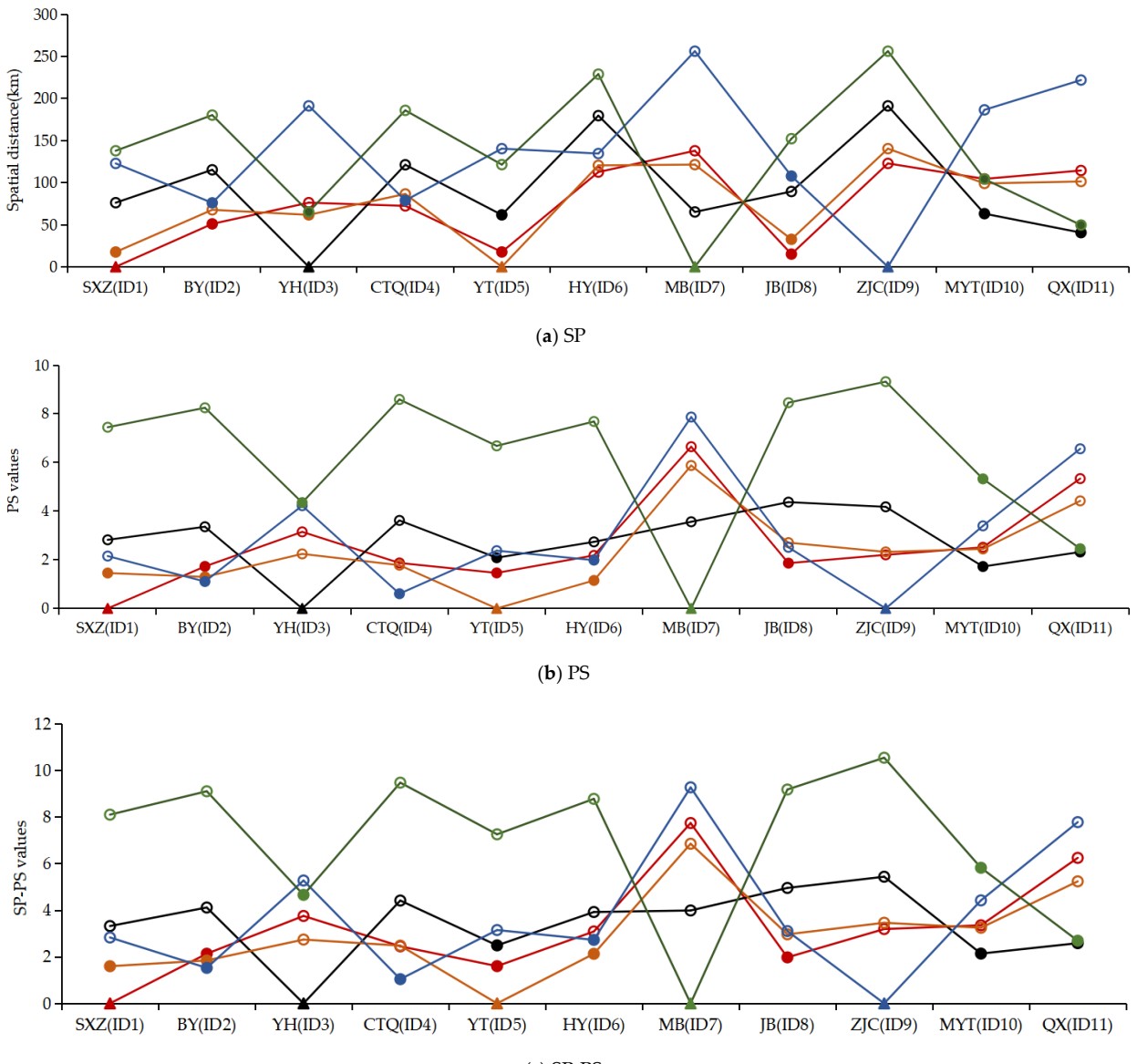

**Figure 4.** Similarity value between reference basin and target basin calculated by (**a**) SP; (**b**) PS; (**c**) SP − PS approach. The ordinates of (**a**–**c**) represent the spatial distance (km) of the center of mass, the physical similarity value, and the integration similarity value between the reference basin and the target basin, respectively. The horizontal axis is the 11 study basins: the solid triangle represents the target basin. The solid circle reflects the three most similar reference basins selected for the corresponding target basin.

**Table 6.** Reference basins selected for each ungauged basin and their corresponding similarity values based on SP, PS, and SP − PS.

| Target Basins | SP | | PS | | SP − PS | |
|---|---|---|---|---|---|---|
| | JB(ID8) | 15.1 | YT(ID5) | 1.46 | YT(ID5) | 1.6 |
| SXZ(ID1) | YT(ID5) | 17.6 | BY(ID2) | 1.73 | JB(ID8) | 1.98 |
| | BY(ID2) | 51 | JB(ID8) | 1.86 | BY(ID2) | 2.14 |
| | QX(ID11) | 40.6 | MYT(ID10) | 1.72 | MYT(ID10) | 2.14 |
| YH(ID3) | YT(ID5) | 61.7 | YT(ID5) | 2.09 | YT(ID5) | 2.49 |
| | MYT(ID10) | 63.1 | QX(ID11) | 2.32 | QX(ID11) | 2.59 |
| | SXZ(ID1) | 17.6 | HY(ID6) | 1.15 | SXZ(ID1) | 1.6 |
| YT(ID5) | JB(ID8) | 32.6 | BY(ID2) | 1.3 | BY(ID2) | 1.85 |
| | YH(ID3) | 61.7 | SXZ(ID1) | 1.46 | HY(ID6) | 2.13 |
| | QX(ID11) | 49.6 | QX(ID11) | 2.46 | QX(ID11) | 2.7 |
| MB(ID7) | YH(ID3) | 65.2 | YH(ID3) | 4.35 | YH(ID3) | 4.67 |
| | MYT(ID10) | 104.2 | MYT(ID10) | 5.33 | MYT(ID10) | 5.83 |
| | BY(ID2) | 76 | CTQ(ID4) | 0.61 | CTQ(ID4) | 1.04 |
| ZJC(ID9) | CTQ(ID4) | 78.9 | BY(ID2) | 1.11 | BY(ID2) | 1.53 |
| | JB(ID8) | 107.8 | HY(ID6) | 1.99 | HY(ID6) | 2.73 |

*4.4. The Results of Regionalizations*

In this study, a total of 12 regionalization approaches were used for each data-deficient basin. Therefore, it is necessary to conduct an overall evaluation of the river discharge simulation results of the regionalization approaches for each ungauged basin. Figure 5 shows the Taylor diagram [58] of each BTOP simulation result using different regionalization approaches for five target basins. It uses three indicators (correlation coefficient (*R*), normalized standard deviation (*NSD*), and normalized root mean square deviation (*NRMSD*)) in one graph to evaluate how close the simulation is to the measured discharge. The *R* shows correlation, *NSD* reflects the degree of dispersion, and *NRMSD* is sensitive to outliers in data series. In general, the river discharge simulation results of all regionalization approaches in the study basin were inferior to the simulation results during the model validation period, except for the YH(ID3). Both the traditional and RA regionalization approaches showed satisfied simulation results in the YH basin, even better than the validated simulation results (as other markers are closer to the reference point (REF) than the black circle in Figure 5b). However, for the regionalization approaches, the traditional approach was superior to the regression-augmented ones. For the SXZ(ID1), YH(ID3), and ZJC(ID9) basins, as shown in Figure 5a,b,e, the triangle is closer to the REF, followed by the square and the five-star icon, indicating that "SP/PS/SP − PS + RI" and "SP/PS/SP − PS + RII" have more significant simulation effects than "SP/PS/SP − PS + RIII". In the same way, for the YT(ID5) and MB(ID7) basins, the simulation effect of "SP/PS/SP − PS + RI" is better, and the simulation effect of "SP/PS/SP − PS + RII" and "SP/PS/SP − PS + RIII" are very close. The evaluation of these regionalizations in each target basin provide some implications for the following regionalization analysis.

Table 7 shows the BTOP model simulation evaluation results of all regionalizations conducted in this study. The application of traditional regionalization approaches in the study area showed that the simulation effect in SXZ(ID1) and YH(ID3) is good. The simulation results were satisfactory in target basin YT(ID5). In watershed MB(ID7), the simulation results were generally good. The traditional regionalization approach had a favorable application effect in the five selected basins. On the whole, among the three traditional regionalization approaches of SP, PS, and SP − PS, SP had the best simulation effect in the research area, which is consistent with Figure 5.

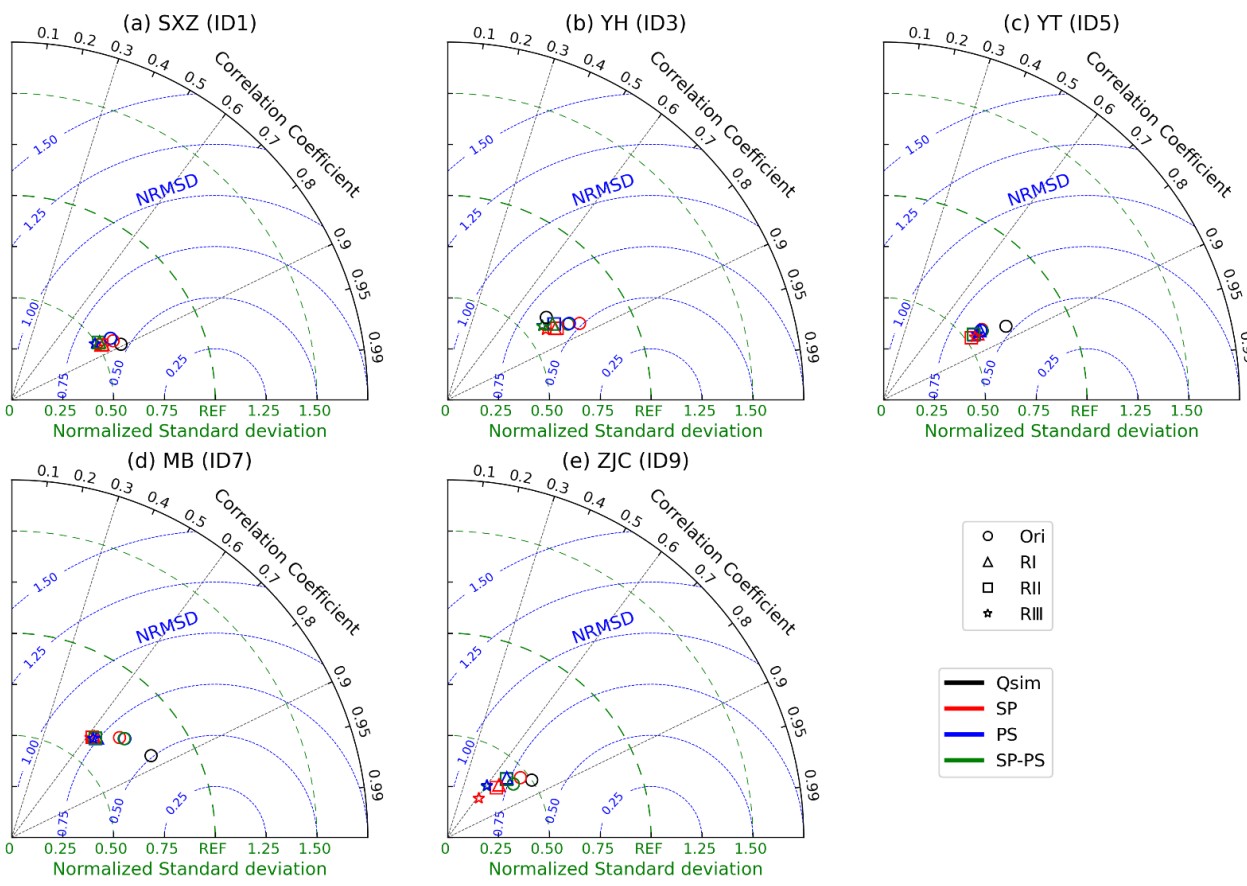

**Figure 5.** Taylor diagram of the simulated results of 5 target basins. (**a**–**e**) Present five ungauged basins, respectively. REF is the observed discharge; Ori represents the original simulated results (by auto-calibrating the model); RI, RII, and RIII are the approaches with regression I, regression II, and regression III, respectively. For example, the red triangle means the river discharge simulation results using the SP + RI approach.

**Table 7.** The river discharge simulation results of each regionalized approach in the target basins.

| Data-Deficient Basin | Evaluation Index | Validation | SP | PS | SP − PS | SP + R I | SP + RII | SP + R III | PS + RI | PS + R II | PS + R III | SP − PS + R I | SP − PS + R II | SP − PS + R III |
|---|---|---|---|---|---|---|---|---|---|---|---|---|---|---|
| SXZ(ID1) | *NSE* | 0.71 | 0.66 | 0.65 | 0.64 | 0.62 | 0.62 | 0.60 | 0.60 | 0.60 | 0.57 | 0.60 | 0.59 | 0.57 |
| | $R^2$ | 0.79 | 0.75 | 0.72 | 0.72 | 0.73 | 0.73 | 0.72 | 0.70 | 0.70 | 0.69 | 0.70 | 0.70 | 0.69 |
| YH(ID3) | *NSE* | 0.55 | 0.71 | 0.67 | 0.67 | 0.63 | 0.64 | 0.59 | 0.61 | 0.61 | 0.56 | 0.60 | 0.61 | 0.56 |
| | $R^2$ | 0.60 | 0.75 | 0.72 | 0.72 | 0.70 | 0.70 | 0.67 | 0.66 | 0.66 | 0.62 | 0.66 | 0.66 | 0.62 |
| YT(ID5) | NSE | 0.71 | 0.61 | 0.61 | 0.62 | 0.61 | 0.58 | 0.60 | 0.61 | 0.59 | 0.60 | 0.60 | 0.58 | 0.60 |
| | $R^2$ | 0.73 | 0.66 | 0.66 | 0.67 | 0.67 | 0.67 | 0.68 | 0.66 | 0.66 | 0.67 | 0.66 | 0.66 | 0.67 |
| MB(ID7) | NSE | 0.71 | 0.51 | 0.54 | 0.53 | 0.38 | 0.37 | 0.36 | 0.40 | 0.39 | 0.38 | 0.40 | 0.39 | 0.38 |
| | $R^2$ | 0.74 | 0.54 | 0.57 | 0.57 | 0.40 | 0.39 | 0.38 | 0.43 | 0.42 | 0.41 | 0.43 | 0.42 | 0.40 |
| ZJC(ID9) | NSE | 0.57 | 0.49 | 0.46 | 0.46 | 0.35 | 0.34 | 0.19 | 0.40 | 0.40 | 0.25 | 0.40 | 0.40 | 0.25 |
| | $R^2$ | 0.68 | 0.60 | 0.60 | 0.60 | 0.49 | 0.49 | 0.39 | 0.50 | 0.51 | 0.37 | 0.50 | 0.51 | 0.37 |

The RA regionalizations generally presented a good simulation effect in SXZ (ID1), YH (ID3), and YT (ID5). However, the results in MB (ID7) and ZJC (ID9) were not ideal. Taking YH (ID3) as an example, the $R^2$ and *NSE* of river discharge simulation results of the SP + RI approach were 0.70 and 0.63, respectively. The $R^2$ and *NSE* of SP + RII were 0.70 and 0.64, respectively, representing an excellent simulation effect and indicating a certain simulation potential. As shown in Figure 6a, the SP performed better, and SP − PS + RIII performed the least, but overall, the difference was not prominent. The river discharge simulation results of the regional approach showed that the low-flow simulation was on the high side, while the high-flow simulation was on the low side, and the difference in simulation results of various regional approaches was mainly reflected in the maximum flow peak simulation.

Figure 6b–d intuitively show the simulation of the RA approaches in the peak flow period of YH (ID3): among the approaches with regression equation, the RII performed the best simulation effect, while RIII received the worst simulation effect. In Table 7, the simulation effect of RI and RII combined with the traditional regionalization approaches were generally better than that of RIII in five study basins. The distinction of RI, RII, and RIII lied in the set of watershed characteristic factors, taken into account by the regression equation in the RA approach, which implies that the watershed characteristic factors set of the model parameter regression equation had a non-negligible influence on the simulation results. Therefore, determining the optimal characteristic factors set in the parameter transfer equation of the BTOP model parameters should be focused on in future research. Figure 6e shows that "SP + RII" had the best simulation effect among "SP/PS/SP − PS + RII". It indicates that the regression-augmented spatial similarity approach could get a good regionalization in some circumstances, which agrees with the findings from Arsenault et al. (2019) [18].

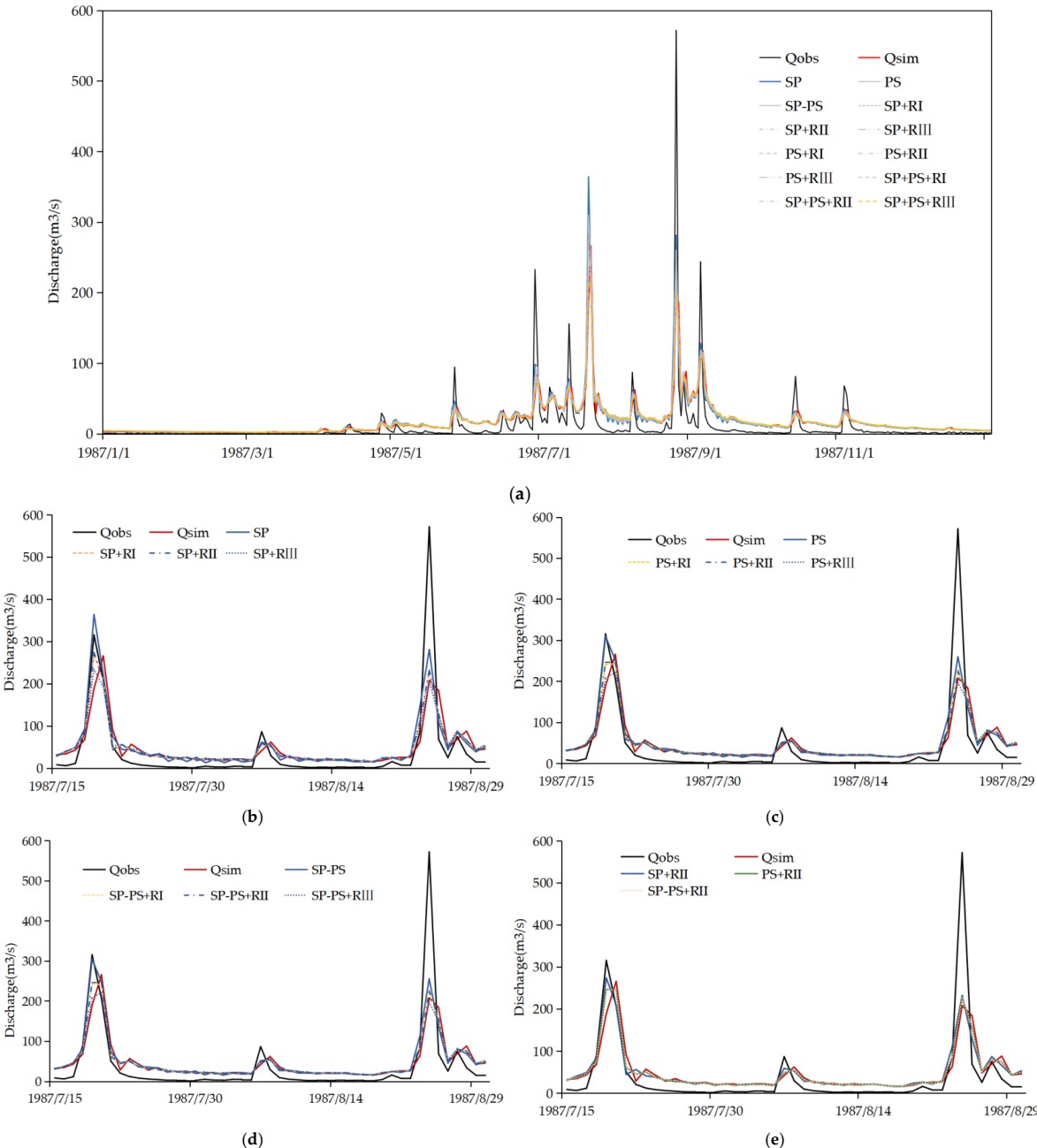

**Figure 6.** Hydrographs of each regionalization approach of basin YH(ID3). (**a**) Daily hydrograph in 1987; (**b**–**d**) represent the hydrograph of RA approaches in the flood period; (**e**) represents a comparison of SP/PS/SP − PS + RII in the flood period.

Moreover, there are three parameters (see Table 5) that were replaced by linear regression in the RA regionalization approaches, accounting for 60% of the calibration parameters of the BTOP model. This regression strength analysis showed that in the RA approach, more than half of the parameters of the BTOP model were obtained from the regression equation. Thus, its simulation performance was directly related to the regression equation, and this analysis leads to additional considerations for the RA approach, in that the regression strength may also be a significant factor affecting the simulation result.

In summary, the five "data-deficient basins" results in this study indicated that the traditional regionalization approach is superior to the RA regionalization approach. We summarized several possible reasons for the outcome. First, due to the limited number of small watersheds, the established relationship between the parameters and the watershed characteristic factors was not accurate enough. Thus, there is still room for improvement in the regression equation, as the regression is vital for the RA approach. Secondly, the regression strength was an influence coefficient for the simulation result of the RA approach. Lastly, compared with previous research which adopts the SP approach [11,18], the small and medium basins in the study area were relatively closely situated. Traditional regionalization approaches are based on the characteristics of watersheds or spatial distances, which makes the traditional regionalization approaches have more advantages than RA. In contrast, the preponderance of RA regionalization approaches have not been highlighted. Introducing more donor basins and establishing representative regression equations that are combined with traditional regionalization approaches are critical to improving the applicability of RA regionalization approaches in ungauged basins.

## 5. Conclusions

Relying on the nine processed watershed characteristic factors (including two new characteristic factors: mean soil particle size ($\Phi$) and mean root depth ($r$)), this study carried out correlation analysis on the five model parameters of the BTOP model. We established the regression equation between the parameters and the selected watershed characteristic factors with 12 regionalization approaches, combining regression equations with traditional regionalization approaches. Overall, we explored the application of the regression-augmented approach for the BTOP model in five small ungauged target basins in the Jialing River Basin, China. The main conclusions are as follows:

(1) This paper initially determined that among the five parameters of the BTOP model, the three parameters $SD_{bar}$, $m$, and $\alpha$ have high correlation ($R > 0.6$) basin characteristic factors, and the $\Phi$ and $r$ both played a significant role in establishing regression equations. We constructed linear regression equations between these three parameters, and the relevant watershed characteristic factors, which could contribute to the establishment of the parameter transfer function of the BTOP model in sparsely gauged regions.

(2) The present work verifies the applicability of the regression-augmented (RA) regionalization approach. In the case of the limited number of the research basins, the RA regionalization approach showed a considerable simulation effect in specific circumstances, indicating that the RA approach has some great potential.

(3) This paper fills the gap in the regionalization research of the BTOP model in the data-deficient basin. It also preliminarily evaluated the application of traditional and RA regionalization approaches, based on BTOP in the ungauged basins. The outcome of this study provides a reference for seeking the optimal BTOP regionalization approach in the target catchment, and a valuable experience for the river discharge simulation of small and medium-sized data-sparse catchments worldwide, especially for some developing countries with limited conditions.

Future work of the regression-augmented regionalization approach of ungauged or sparsely gauged basins, to establish a more representative regression equation and find a regression equation that better combines with the traditional regionalization approach, should focus on the following aspects: introducing more donor basins, classifying the participating basins in advance, increasing the number of watershed characteristic factors,

and establishing the regression equation by determining the optimal set of watershed characteristic factors for each sensitive parameter.

**Author Contributions:** T.A., L.Z. and Y.Z. designed this research; Y.Z., L.L., F.Q., L.Z., X.Z., T.C. and X.L. collected the data; Y.Z., L.L. and F.Q. analyzed the data and wrote the draft. L.Z. and T.A. revised the manuscript. All authors have read and agreed to the published version of the manuscript.

**Funding:** We gratefully acknowledge the Regional Innovation Cooperation Program (2020YFQ0013) and Key R&D Project (2021YFS028) from the Science and Technology Department of Sichuan Province, China Scholarship Council (201906240035).

**Institutional Review Board Statement:** Not applicable.

**Informed Consent Statement:** Not applicable.

**Data Availability Statement:** The study did not report any data.

**Acknowledgments:** The authors thank the anonymous referees for their valuable comments and suggestions.

**Conflicts of Interest:** The authors declare no conflict of interest.

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
