# Peer review of "Application of the Regression-Augmented Regionalization Approach for BTOP Model in Ungauged Basins"

_water, doi:10.3390/w13162294_

Round 1

Reviewer 1 Report

I have read the paper and conclude that it can be published in its present form. There is a good agreement of the model simulations concerning the observations, showing the accuracy of BTOP.

Please find attached the specific suggestions.

Reviewer 2 Report

Thank you for the opportunity to review the manuscript "Application of the Regression-augmented Regionalization Approach for BTOP Model in Ungauged Basins" written by Zhou et al. The authors conducted an interesting investigation for transferrability of model parameters. Despite a small number of catchments (11 with 5 randomly selected) and close proximity of the study catchments, this study is of great interest to the scientific community. The manuscript is relatively well-written with good quality of figures, tables and references. However, I found many confusing statements and suggested revision to improve clarity and readibility in the revised version. Therefore, I suggest to accept this manuscript with minor revisions prior publication the Journal. I enjoyed reading the manuscript and marked my major concerns and minor comments in the pdf file, see attached document.
